# Radiolabeling, Quality Control and In Vivo Imaging of Multimodal Targeted Nanomedicines

**DOI:** 10.3390/pharmaceutics14122679

**Published:** 2022-12-01

**Authors:** Phuoc-Vinh Nguyen, Emilie Allard-Vannier, Nicolas Aubrey, Christine Labrugère-Sarroste, Igor Chourpa, Julien Sobilo, Alain Le Pape, Katel Hervé-Aubert

**Affiliations:** 1EA6295 Nanomedicines and Nanoprobes, University of Tours, 37000 Tours, France; 2School of Medicine, Vietnam National University Ho Chi Minh City, Ho Chi Minh City 700000, Vietnam; 3ISP UMR 1282, INRA, BioMAP, University of Tours, 37000 Tours, France; 4PLACAMAT UMS 3625, CNRS, University of Bordeaux, 33600 Pessac, France; 5CIPA, TAAM CNRS, 45000 Orléans, France

**Keywords:** EGFR, scFv, nanomedicine, in vivo tracking, SPECT-CT, radiolabeling

## Abstract

Following our previous study on the development of EGFR-targeted nanomedicine (NM-scFv) for the active delivery of siRNA in EGFR-positive cancers, this study focuses on the development and the quality control of a radiolabeling method to track it in in vivo conditions with nuclear imaging. Our NM-scFv is based on the electrostatic complexation of targeted nanovector (NV-scFv), siRNA and two cationic polymers. NV-scFv comprises an inorganic core, a fluorescent dye, a polymer layer and anti-EGFR ligands. To track NM-scFv in vivo with nuclear imaging, the DTPA chemistry was used to radiolabel NM-scFv with ^111^In. DTPA was thiolated and introduced onto NV-scFv via the maleimide chemistry. To obtain suitable radiolabeling efficiency, different DTPA/NV-scFv ratios were tested, including 0.03, 0.3 and 0.6. At the optimized ratio (where the DTPA/NV-scFv ratio was 0.3), a high radiolabeling yield was achieved (98%) and neither DTPA-derivatization nor indium-radiolabeling showed any impact on NM-scFv’s physicochemical characteristics (D_H_ ~100 nm, PDi < 0.24). The selected NM-scFv-DTPA demonstrated good siRNA protection capacity and comparable in vitro transfection efficiency into EGFR-overexpressing cells in comparison to that of non-derivatized NM-scFv (around 67%). Eventually, it was able to track both qualitatively and quantitatively NM-scFv in in vivo environments with nuclear imaging. Both the radiolabeling and the NM-scFv showed a high in vivo stability level. Altogether, a radiolabeling method using DTPA chemistry was developed with success in this study to track our NM-scFv in in vivo conditions without any impact on its active targeting and physicochemical properties, highlighting the potential of our NM-scFv for future theranostic applications in EGFR-overexpressing cancers.

## 1. Introduction

Recently, theranostic strategies have been extensively exploited in oncology for the combination of simultaneous detection and killing of various types of tumors. The term “theranostic” was introduced in 2002 and refers to the design of platforms for both therapeutic and diagnostic functions [1]. On the one hand, these systems can exert antitumor activities through their encapsulated therapeutic agents, e.g., drugs/genes/photosensitizers. On the other hand, theranostic tools can help track the in vivo journey of therapeutic agents as regards their pharmacokinetics/biodistribution and monitor treatment efficacy by delivering magnetic contrast/fluorescent/radioactive agents [2,3]. Among available systems, nanoparticles (NPs) hold great potential to revolutionize the future of cancer management. To this end, an ideal theranostic system must meet several requirements, e.g., (i) safety, (ii) rapid and selective tumor accumulation, (iii) ability to report biochemical and morphological tumor characteristics, (iv) efficient delivery of a sufficient amount of “drugs” without damaging healthy organs, and (v) clearance from the body within hours or biodegradation into nontoxic by-products [4,5]. So far, most theranostic NPs have focused on passive targeting strategies relying on the permeability and retention effect (EPR) [2,6]. However, the transition of such nanodrugs from benchtop to clinic remains difficult [2]. Besides, the functionalization of these NPs with biological ligands, which target specific molecules or receptors present on cancer cell surfaces can bring additional advantages [7,8,9].

In accordance with the current tendency towards targeted therapy for cancer treatment, gene therapy has emerged as an innovative modality for cancer treatment and is drawing great attention in specific cancer therapy [10]. Many recent studies have demonstrated the potential of this strategy in improving the antitumor activities of chemotherapeutic agents via its effect on oncoproteins’ downregulation [10,11]. In gene therapy using interfering RNAs, three RNA types have been developed and largely exploited, e.g., small hairpin RNA (shRNA), micro-RNA (miRNA) and small interfering RNA (siRNA). Their mechanism is based on the activity of double-stranded RNA on the expression of a particular gene with a homologous sequence [12,13,14]. Due to their high specificity, these RNAis are able to inhibit tumor growth via specific oncogenes’ downregulation if they are delivered into cancer cells. Furthermore, their easy synthesis and negligible side effects make gene therapy one of the most promising strategies in cancer management [12,13]. Despite its therapeutic potency, its degradation by nucleases and hard internalization across cellular membranes are currently restricting the application of these RNAis. To address these issues, nanomedicines (NMs) are among the most prospective candidates [12,13]. In our previous studies, a targeted nanomedicine for siRNA delivery was successfully developed and proved its in vitro potency for actively targeting EGFR-positive cancers [11]. However, before evaluating its in vivo anticancer activity and its safety, it is required to develop and validate a reliable imaging method for tracking them qualitatively and quantitatively in vivo which is also the main objective of the current study.

For cancer imaging with NMs, three types of modalities are available, including magnetic resonance (MRI), optical and nuclear imaging [15,16]. Among them, MRI is based on the use of innately magnetic NPs such as SPIONs and optical imaging refers to the incorporation of a fluorescent dye onto NPs or the use of innately fluorescent materials such as quantum dots to construct NPs [17,18]. The last type of imaging is nuclear imaging, which comprises the functionalization of NMs with radionuclides for molecular imaging such as positron emission tomography (PET) and single-photon emission computed tomography (SPECT) [16]. For optical imaging, near-infrared fluorescence (NIRF) is favored because of its remarkable signal-to-noise ratios and its low tissue absorption in the spectral range between 650 and 900 nm [15]. However, NIRF imaging is limited due to its low in vivo stability [19]. In another approach, while NIRF imaging gives only qualitative information, nuclear imaging presents an important advantage as it provides a quantitative and high level of resolution [16]. SPECT is a well-established imaging modality that uses signals emitted by gamma-emitting radiotracers to construct images. Recent advances in hardware scanner technology combining SPECT and CT (computed X-tomography) have made it possible to build imaging devices with higher resolutions and to improve the anatomic location, enabling us to track both qualitatively and quantitatively radiolabeled NPs in in vivo conditions [20,21]. For the application of SPECT, a gamma-emitting radiotracer such as iodine-123, technetium-99m, xenon-133, thallium-201, and indium-111 is needed [22]. Among these radiotracers, indium-111 presents many advantages, especially in terms of physical half-life time. Indeed, the half-life time of Indium-111 is up to 67 h allowing the possibility of evaluating treatment schemes evaluation and assessing long-term biodistribution [23,24]. Furthermore, an additional therapeutic effect can be also obtained with indium-111 via Auger electrons emitted [24]. Last but not least, the possibility of combining two or even more isotopes (e.g., indium-111 and technetium-99m) can allow greater anatomic delineation of tumor location and makes Indium-111 is one of the most widely used radiotracers for SPECT in the clinic [23]. Due to its benefits, indium-111 is currently conjugated to peptides, monoclonal antibodies and nanoparticles for not only tumor imaging but also image-guided surgery or post-radical prostatectomy management for a wide range of cancers (breast, prostate, colorectal, and lung cancers, etc.) [24,25,26,27].

In this study, a radiolabeling method for our EGFR-targeted nanomedicine (NM-scFv) was developed and controlled in terms of its quality both in in vitro and in in vivo conditions. Our NM-scFv is based on the complexation between siRNA, two cationic polymers and targeted nanovectors (NV-scFv) made of superparamagnetic nanoparticles (SPIONs). On the inorganic core of NV-scFv, a NIRF dye was directly introduced and embedded under a polymer. These NPs were functionalized with anti-EGFR scFv ligands for active targeting by grafting them onto the extremity of the polymer layer via the maleimide chemistry. To introduce a radiotracer onto NPs, diethylenetriaminepentacetic acid (DTPA) was used for ^111^In SPECT-CT imaging. To this end, DTPA was thiolated and conjugated onto NPs using the maleimide chemistry. Three batches of NPs corresponding to three molar DTPA/NV ratios were tested. The resulting NPs were evaluated in terms of physicochemical properties, siRNA protection capacity, and the yield of radiolabeling to figure out the most prominent candidate. The active siRNA delivery of the DTPA-derivatized NM-scFv was evaluated in vitro with EGFR-overexpressing cancer cells and compared to non-derivatized NM-scFv. Besides, the stability in in vivo conditions of the complexation between DTPA-derivatized NV-scFv and siRNA was also studied using NIRF imaging. In the end, SPECT-CT imaging was applied to assess the efficiency of our radiolabeling method in in vivo experiments.

## 2. Materials and Methods

### 2.1. Materials

Diethylenetriaminepentacetic acid bis-anhydride (DTPA-BA), dimethylformamide (DMF), aminoethanethiol, trimethylamine, chitosan (MW = 110–150 kDa; the degree of acetylation ≤40 mol.%) and poly-L-arginine (PLR, MW 15–70 kDa), disposable PD 10 desalting columns were purchased from Sigma-Aldrich Chimie GmbH, St. Quentin Fallavier, France. N-Hydroxysuccinimide-Polyethylene glycol 5000-Maleimide (NHS-PEG_5000_-Maleimide) was obtained from Rapp Polymer, Tübingen, Germany. Dylight^TM^680 NHS ester-reactive dye, Coomassie Plus Assay Kit, protein L-peroxidase, and heparin were from Thermo Scientific, Rockford, IL, USA. Control siRNA, anti-luciferase siRNA and control siRNA labeled with Atto^TM^740 were provided by Ambion^®^, New York, NY, USA. 3,3′,5,5′-Tetramethylbenzidine substrate (TMB), fetal bovine serum (FBS) was purchased from Sigma, St. Louis, MO, USA. Agarose gel loading dye 6× was from Fisher, Bioreagent^®^, Illkirch, France. Epidermal growth factor receptor (EGFR) recombinant protein was provided by Sino Biologicals, Beijing, China. RPMI-1640 medium, L-glutamine 200 mM, penicillin/streptomycin and trypsin were obtained from Gibco^®^, Life Technologies, Paisley, UK. Humanized anti-EGFR single-chain variable fragment (scFv) fragments were designed and synthesized by the BioMAP team, ISP UMR 1282, INRA, University of Tours, Tours, France. indium-111 chloride (^111^InCl_3_) was obtained from CIS Bio International, Gif-sur-Yvette, France and indium-115 chloride (^115^InCl_3_) was purchased from Sigma-Aldrich Chimie GmbH, St. Quentin Fallavier, France. All other chemicals were analytical reagent grade.

### 2.2. Cell Culture

Luciferase stably expressing human non-small cell lung cancer (H460-Luc) was obtained from Caliper Life Sciences Inc. (Hopkinton, MA, USA). Cells were grown at 37 °C in an atmosphere containing 5% of CO_2_. The culture medium was made of the RPMI-1640 medium supplemented with 10% of fetal bovine serum (FBS), 2 mM of L-glutamine, and 1% of penicillin/streptomycin. The cell harvesting was made with trypsin/EDTA (0.05%) at 80% of confluence.

### 2.3. Preparation of DTPA-Derivatized EGFR-Targeted Nanomedicine (NM-scFv-DTPA)

The preparation of the NM-scFv-DTPA comprises three main steps including (i) the thiolation of DTPA-BA, (ii) preparation of NV-scFv and DTPA-derivatization, and (iii) NM-scFv-DTPA formulation.

#### 2.3.1. Thiolation of DTPA-BA

(a)Synthesis of the thiolated DTPA (DTPA-SH)

In a round flask of 100 mL, 0.5 g (1.4 × 10^−3^ mol) of diethylenetriaminepentacetic acid bis-anhydride (DTPA-BA) was dissolved in 10 mL of dimethylformamide (DMF) and then heated up to 70 °C. In a second flask, 0.22 g (2.8 × 10^−3^ mol) of aminoethanethiol was dissolved in 7.5 mL of DMF and 0.435 mL of trimethylamine. The solution was added to the round flask of DTPA-BA and the reaction took place overnight at 70 °C under magnetic stirring. The obtained solution was cooled down to room temperature and precipitated drop by drop in a cold chloroform solution. A white powder was obtained, filtered, washed with a chloroform solution and dried under a vacuum. The DTPA-SH was obtained and stored in inert gas at −20 °C.

(b)Characterization of the thiolated DTPA

To verify the presence of thiol groups on DTPA-SH, several analytical methods were applied, including infrared spectroscopy (IR) and RMN analysis. For IR analysis, 10 mg powder of the obtained DTPA-SH was dissolved in 1 mL of distilled water and the IR spectrum was recorded with IR spectroscopy (Bruker Vector 22, ATR mode, Billerica, MA, USA). ^1^H-NMR spectra were recorded with a Bruker 300 MHz spectrometer at 25 °C with deuterium oxide (D_2_O) as solvent.

#### 2.3.2. Preparation and Characterization of NV-scFv and DTPA-Derivatization

(a)Preparation of NV-scFv

The preparation of the NV-scFv is based on the previous protocol developed in our group and includes three steps as follows: (i) coupling of NIRF dye onto silanized SPION surface, (ii) PEGylation with NHS-PEG_5000_-maleimide and (iii) bioconjugation with the humanized anti-EGFR scFv ligands [12].

(b)DTPA-derivatization of the NV-scFv (NV-scFv-DTPA)

The introduction of DTPA-SH onto the NV-scFv was performed via the reaction between the thiol functional groups on DTPA-SH and the remaining maleimide functions on NV-scFv after the bioconjugation of NVs with humanized scFv. To express the ratio between NV-scFv and thiolated DTPA, the molar ratio of DTPA/Fe was used. Three molar ratios (DTPA/Fe) were tested including 0.03, 0.3, and 0.6 to find out the most suitable candidate for SPECT-CT imaging. Several quantities of DTPA-SH, each corresponding to a quantity of NV-scFv, were dissolved in PBS 1X and then added into NV-scFv. The reaction lasted two hours, under agitation and at room temperature. Afterward, the extensive maleimide groups were saturated by the addition of CH_3_O-PEG_600_-SH. As a control, NV-scFv without DTPA was synthesized in the same conditions.

NV-scFv and NV-scFv-DTPA were collected and purified by a size-exclusion chromatography (SEC) system using the AKTA purifier FPLC system equipped with a prepacked Superdex 200 pg column (600 × 16 mm^2^) (GE Healthcare BioScience AB, Uppsala, Sweden) and a PBS 1X solution as mobile phase (flow rate of 1.6 mL/min). At the end of the purification, NV-scFv-DTPA was collected and re-concentrated with Vivaspin^®^ (cut-off 30 kDa, Fisher Scientific, Illkrich, France).

(c)Characterizations of NV-scFv-DTPA

X-ray Photoelectron Spectroscopy (XPS) Characterization

The concentration of NV-scFv and NV-scFv-DTPA_0.6_ (DTPA/Fe ratio at 0.6) was adjusted to have the same concentration of grafted scFv ligands at 17.56 µM of scFv. A Thermo Fisher Scientific K-ALPHA spectrometer was used for XPS surface analysis with a monochromatized AlKα source (hν = 1486.6 eV) and an X-Ray spot at 400 microns in diameter. Sample preparation consisted of depositing several drops of NVs suspension onto an indium foil. The sample holder was directly put into a vacuum and a pressure of 10^−5^ Pa was reached before being transferred into the analysis chamber. The full spectra (0–1100 eV) were acquired with a constant pass energy of 200 eV and high-resolution spectra were obtained with a constant pass energy of 40 eV. Charge neutralization was applied during the analysis. High-resolution spectra (i.e., C1s, O1s, N1s, Fe2p) were quantified (Scofield sensitivity factors applied) and/or fitted using the AVANTAGE software provided by ThermoFisher Scientific.

Size and Zeta Potential Characterizations

The hydrodynamic diameter (D_H_), the polydispersity index (PDi), and the zeta potential (ζ) were determined using Nanosizer apparatus (Zetasizer^®^, Malvern Instrument, Malvern, UK) in PBS 1X for D_H_ or NaCl 0.01 M for ζ at the concentration of 50 mg of iron/L. The D_H_ was based on intensity. All the measurements were achieved at 25 °C in triplicate and presented in mean values ± standard deviation.

Quantification of Conjugated Antibody Fragments

The quantity of conjugated anti-EGFR scFv ligands was determined by a modified Bradford assay as detailed in our previous publication [12]. Briefly, the samples were incubated with the reactive agent and the absorbance was determined at 630 nm. The concentration of conjugated scFv was determined and the number of scFv per NV-scFV-DTPA was estimated.

#### 2.3.3. Preparation and Characterization of EGFR-Targeted Nanomedicine (NM-scFv-DTPA)

The formulation of NM-scFv including NV-scFv-DTPA, siRNA, chitosan and poly-L-arginine was performed according to our previously published article using the electrostatic interactions [12]. To determine the content, and the mass ratio of NV-scFv/siRNA (MS) was used to determine the quantity of NV-scFv and the charge ratio of chitosan/siRNA (CS) and PLR/siRNA (CR) was used for cationic polymers. All these ratios were previously optimized and determined at 10 for optimal gene delivery [12]. The concentration of the finalized NM-scFv could be expressed in two ways, via either the concentration of complexed siRNA (nM), or the concentration of iron in the formulation (mg of iron/L).

The D_H_ of all batches of NM-scFv-DTPA was measured in distilled water at the concentration of 50 mg of iron/L and the ζ was determined in NaNO_3_ 0.01 M (pH = 7.6) at the concentration of 6.25 mg of iron/L.

### 2.4. In Vitro Evaluation of the Impact of DTPA-Coupling on NM-scFv Active Targeting Properties

#### 2.4.1. Functionality Test of Grafted Antibody Fragments on NM-scFv-DTPA

Enzyme-linked immunosorbent assay (ELISA) was used to evaluate the functionality of grafted scFv fragments on NM-scFv-DTPA towards its target, the EGFR protein. To this aim, EGFR recombinant in PBS 1X at a concentration of 2 µg/mL was first coated in a 96-well plate and incubated for 16 h at 4 °C. Afterward, BSA 3% in PBS 1X was used to saturate the coated wells during 1 h at 37 °C. PBS for the negative control, NM, NM-scFv, or NM-scFv-DTPA (from 0.003 to 100 mg of iron/L) were added in wells and incubated at 37 °C for 1 h. The washing solution made of PBS-Tween 20 (0.05%) was then used to rinse the wells and they were finally incubated for 1 h at 37 °C with protein L-peroxidase at a concentration of 1.25 µg/mL. To reveal the presence as well as the content of scFv in each batch of nanosystem, an enzymatic reaction was applied by the addition of TMB reactive and stopped with H_2_SO_4_ 1M. The absorbance of the final solution in wells was determined at 450 nm using a microplate reader (Bio-Tek^®^, Inc., Winooski, VT, USA).

#### 2.4.2. Transfection Assay

H460-Luc cells were seeded at 2500 cells/well in a 96-well plate for 24 h before the transfection. On the day of transfection, NM, NM-scFv, NM-scFv-DTPA and Lipofectamine^TM^RNAimax (Lipofectamine) were prepared with anti-luciferase siRNA at a final concentration of 100 nM according to the formulation protocol and the manufacturer’s recommendation. Cells were treated with each kind of nanomedicine in serum-free Opti-MEM for 4 h. After 4 h, a culture medium enriched in 20% of FBS was added and maintained for 44 h until the analysis. Non-treated cells and siRNA formulated with Lipofectamine were used as the negative and positive controls, respectively. After 48 h of incubation, the medium was removed and washed twice with HBSS buffer and replaced by a fresh culture medium. The equivalent volume of ONE-Glo^TM^ reagent (Promega, The Netherlands) was added. The intensity of the luminescence signal was measured with the microplate reader Synergy H1 (Biotek, Colmar, France). The transfection efficiency was calculated according to the following formula:$$\mathrm{Transfection}\;\mathrm{efficiency}\;(\%)=100\times \frac{100\times RLU_{sample}}{RLU_{control}}$$
where: *RLU_sample_* is the average luminescence intensity of the samples (NMs or siRNA-Lipofectamine), *RLU_control_* is that of non-treated cells.

All experiments were triplicated and a Student’s *t*-test was used to compare the means of transfection efficiency of each kind of nanomedicine.

### 2.5. Physico-Chemical Properties and siRNA Protection Capacity of NM-scFv-DTPA-^115^In^3+^

To control the physicochemical properties and siRNA protection capacity of the NM-scFv-DTPA before IV injection, a “cold kit” of non-radioactive ^115^InCl_3_ was complexed with tested NM-scFv-DTPA at the same concentration and under identical conditions to their counterparts for in vivo imaging.

#### 2.5.1. Preparation and Physico-Chemical Characterizations of NM-scFv-DTPA-^115^In

Three hundred microliters of solution ^115^InCl_3_ in HCl 0.04 M (0.1 µg/L) at pH 1.6 was firstly neutralized with 300 µL of CH_3_COONa 1 M (pH 8.7) to have an intermediate pH around 5.7. Afterward, 300 µL of the mixture was added to 300 µL of NM-scFv-DTPA synthesized with different DTPA/Fe ratios at 15,000 nM in siRNA (or 1995 mg of iron/L) and the complexation took place for an hour at room temperature. After the complexation, each resulting NM-scFv-DTPA-^115^In was purified through size exclusion chromatography on a PD-10 column eluted with the PBS 1X buffer (pH 7.4). All the fractions (five droplets corresponding to 180 µL per fraction) with the presence of NMs were collected and pooled together. Finally, the D_H_ and ζ of different purified NM-scFv-DTPA-^115^In at the concentration of 50 mg of iron/L in PBS 1× were determined and compared.

#### 2.5.2. siRNA Protection Capacity of NM-scFv-DTPA-^115^In

The electrophoresis technique on agarose gel was employed to verify the siRNA protection capacity of the optimized NM-scFv-DTPA-^115^In purified with the column PD-10. Briefly, an agarose gel at 1% (*m*/*v*) was prepared containing 0.01% (*v*/*v*) ethidium bromide (EtBr) to visualize free siRNA. NM-scFv-DTPA-^115^In was prepared in double at the concentration of 237 nM in siRNA. Two kinds of sample were prepared using two dilution solutions at a dilution factor of 10:1 (*v*/*v*), including distilled water or heparin 10 g/L to destabilize the formulation leading to a release of formulated siRNA. Prior to the electrophoresis step, a loading buffer was added and the final sample at 2.4 pmol of siRNA was deposited in wells. The migration on the gel in a Tris-acetate-EDTA (TAE) 1X buffer of samples was performed for 15 min at 150 V. The presence of free siRNA was visualized with UV-imaging using the EvolutionCapt software on a Fusion Solo.65.WL imager (Vilber Lourmat, Marne-la-Vallée, France).

### 2.6. Radiolabeling Efficiency for Determination of the Suitable DTPA/Fe for In Vivo SPECT-CT Imaging

Two batches of NM-scFv-DTPA at DTPA/Fe = 0.03 and 0.3 were complexed with ^111^InCl_3_ at radioactivity of 138.26 ± 9.51 MBq and evaluated in terms of radiolabeling efficiency. Radiometal labeling efficiency was determined by instant thin-layer chromatography performed on silica-gel strips (ITLC-SG, Pall corporation, East Hills, NY, USA), using 10% (*w*/*v*) ammonium acetate:methanol (1:1) as the mobile phase. A Retention factor (Rf) for “free” radionuclides at 0.8–1.0 and that for radiolabeled NMs and radiocolloids at 0.0–0.3 were applied. Radioactivity distribution was analyzed by electronic autoradiography (Cyclone Plus Phosphor Imager; PerkinElmer, Waltham, MA, USA). The radiolabeling efficiency was calculated using the following formula:$$\mathrm{Yield}\;\mathrm{of}\;\mathrm{radiolabeling}=100\;\times \;\frac{R_{NMs}}{R_{NMs}+R_{free}}$$
where: *R_NMs_* is the radioactivity of radiolabeled NMs (Rf = 0.0–0.3), *R_free_* is the radioactivity of free radionuclides (Rf = 0.8–1.0).

### 2.7. Small-Animal In Vivo Experiments

Female Balb/C nude mice (three animals per group for two groups) were purchased from Janvier Laboratories with a weight of around 20 g/individual (7 weeks). During the acquisition of in vivo imaging, the animals were anesthetized by a mixture of isoflurane/air (Iso-Vet, Piramal Healthcare, Mumbai, India) at 5% (*v*/*v*) and maintained at 2.5% (*v*/*v*). All experiments on animals were performed in accordance with institutional guidelines for animal use specified by the French Ministry of Higher Education, Research and Innovation (APAFIS#29222-2020121807569293 v6).

#### 2.7.1. Radiolabeling of NM-scFv-DTPA with ^111^InCl_3_

The protocol of complexation between NM-scFv-DTPA and ^111^InCl_3_ was identical to that described in Section 2.5.1 with initial radioactivity of 138.26 ± 9.51 MBq. After the purification, as NMs were fluorescent-labeled, the elution profile of purified samples was established by measuring the NIRF intensity of each fraction. Besides, the radioactive elution profile of the purified samples was also established by measuring the radioactivity of each fraction with an activity meter (Capintec CRC-15R, Florham Park, NJ, USA). The six most concentrated fractions were mixed together, controlled in terms of pH, sterilized through a polyvinylidene fluoride membrane of 0.22 µm Sterile Millex^®^ Filter Unit (Merck Millipore, Carrigtwohill, Ireland) and used for IV injection into mice.

To determine the actual concentration of injected radiolabeled NM-scFv after the membrane sterilization, a calibration curve of NM-scFv-DTPA from 79.8 to 1.25 mg of iron/L according to its Dylight^TM^680 NIRF intensity was established beforehand. The NIRF of the radiolabeled NMs used for IV injection was measured and the concentrations of the injected samples were calculated.

#### 2.7.2. Stability Control of Optimized NM-scFv-DTPA-^111^In in In Vivo Conditions

The mice were anesthetized with isoflurane/air at 5% (*v*/*v*) and intravenously (IV) injected with radiolabeled and DTPA-derivatized NM-scFv made of control siRNA fluorescent-labeled with Atto^TM^740, at the dose of 152.5 µg of siRNA per kg of mouse. At 40 min after injection, the mice were sacrificed and subjected to ex vivo NIRF imaging. NIRF images were acquired on a LUMINA II imager (Perkin Elmer, Villebon-sur-Yvette, France). The couples of excitation and emission filters at 675 nm (30 nm bq)/720 nm (20 nm bq) and 745 nm (30 nm bq)/800 (20 nm bq) were used for the detection of Dylight^TM^680 and Atto^TM^740, respectively. The exposure time was 20 s and the binning was set at 4.

#### 2.7.3. Small-Animal SPECT-CT Imaging

The ^111^In-labeled NM-scFv was IV injected into three mice at a dose of 152.5 µg of siRNA per kg of mouse at 3.7 ± 0.74 MBq of radioactivity. At 40 min post-injection, the mice were sacrificed and SPECT-CT images were acquired with a NanoSPECT/CT (Mediso, Budapest, Hungary). Helical SPECT scans with 24 projections of 60 s were acquired before the CT scans with the following parameters: exposure time at 500 m, 55 KVp, 145 µA, 180 projections and pitch set to 1. The reconstruction of SPECT images was performed on the HiSPECT NG software (Scivis GmbH, Göttingen, Germany) and that of CT images was achieved on InVivoScope software (Invicro, Boston, MA, USA). The accumulation of radioactivity in major organs was also quantified using VivoQuant 4.0 (Invicro, Boston, MA, USA) thanks to a mouse phantom filled with known ^111^In activity. The final results were presented as mean values ± standard deviation.

## 3. Results and Discussion

### 3.1. Synthesis and Characterization of NV-scFv-DTPA

To obtain NV-scFv-DTPA, we started with the synthesis of NV-scFv according to the protocol clearly described in our previous publication [12]. Briefly, the silanized SPIONs were firstly functionalized with an NIRF dye of Dylight^TM^680 via NHS chemistry. The fluorescent-labeled SPION surface was then covered by a polymer layer of NHS-PEG_5000_-Maleimide using NHS chemistry. Dylight^TM^680 was directly grafted onto the SPION surface and embedded under the polymer layer, which would help improve its in vivo stability. In addition, to increase the stability and the stealthiness of the resulting NPs, the maleimide functions on the PEG layer allowed further functionalization of NPs with the humanized anti-EGFR scFv by forming the thioether linkage.

Until this step, we obtained our EGFR-targeted NV-scFv. For in vivo tracking with SPECT-CT imaging, the NV-scFv needs to be radiolabeled with radiometal nuclides such as ^99m^Tc or ^111^In. To this end, an intermediate chelator is necessary. Among different available chelators for radiolabeling, DTPA was chosen due to its mild conditions of reaction (at room temperature) and the high stability in the complexation with the radionuclides [28]. For satisfactory radiolabeling, it is required that radionuclides can easily access their chelators. Therefore, DTPA was conjugated directly onto the extremity of the polymer layer. Moreover, the DTPA-derivatization was performed at the last step of synthesis to minimize their possible impact on the scFv coupling, which can lead to a reduction in the active targeting of the finalized NPs. To do so, DTPA was firstly chemically modified by grafting sulfhydryl (SH) groups, and then conjugated onto NV-scFv via maleimide chemistry.

#### 3.1.1. Synthesis and Characterization of DTPA-SH

DTPA was thiolated following a protocol previously reported by Alric et al. [29]. To be more specific, it was the reaction between cysteamine and bis-anhydride DTPA (DTPA-BA) with a molar ratio of 2:1, respectively, in the presence of triethylamine (Figure 1). In order to avoid the hydrolysis of anhydride, the reaction was performed under an inert atmosphere (nitrogen) and in anhydrous DMF.

After the reaction, a white powder soluble in water and in hydro-alcoholic solutions was obtained with a similar yield (around 90%) to that described in the study of Alric et al. [29]. The introduction of thiol groups onto DTPA-BA was verified with IR spectrometry and ^1^H-NMR. Similar to the work of Alric et al., in the FT-IR spectrum, compared to DTPA-BA, the appearance of the characteristic vibrational elongational band of the thiol (SH) function at ~2520 cm^−1^ was recorded, revealing a successful introduction of thiol groups onto DTPA. In terms of NMR analysis, the ^1^H-NMR in D_2_O of the obtained DTPA-SH confirmed the structure of the molecule [29]. The presence of residual DMF was expected due to its high boiling temperature (Appendix A). However, it would be subsequently eliminated during the preparation of NM-scFv with different methods of purification.

#### 3.1.2. DTPA Functionalization of NV-scFv

The next step was to conjugate DTPA-SH onto NPs via the maleimide chemistry between functional SH groups on DTPA-SH and the remaining maleimide groups on the polymer layer of our NV-scFv (Figure 2). Indeed, the total number of maleimide groups present on our NV was identical to that of the PEG molecule which has been found to be around 873 ± 4 molecules of PEG/NV (Vinh Nguyen et al. [9]). After the scFv conjugation step, as the conjugation of one scFv requires only one maleimide functional group, many maleimide groups are available allowing the decoration of our NV-scFv with other molecules possessing suitable functional groups.

To obtain relevant radiolabeling efficiency, a sufficient quantity of grafted DTPA on NV-scFv is of paramount importance. Nevertheless, an overwhelming quantity of grafted DTPA may change the colloidal properties and decrease the stability of the finalized NV-scFv. As a result, the choice of an appropriate DTPA quantity used in this synthesis is essential. To do so, three different molar DTPA/Fe ratios at 0.03, 0.3, and 0.6 were used to synthesize the three corresponding NV-scFv-DTPA_0.03_, NV-scFv-DTPA_0.3_, and NV-scFv-DTPA_0.6_, respectively. The start ratio of DTPA/Fe was chosen at 0.03 due to our calculation of the smallest amount of DTPA to fulfill theoretically all the remaining maleimide functional groups after the scFv conjugation step. Furthermore, to verify if our reaction was successful and DTPA was grafted onto our NV-scFv, the NV-scFv-DTPA_0.6_ and the NV-scFv were assessed and compared by XPS analysis. Besides, different NV-scFv-DTPA were compared to each other and to non-DTPA-derivatized NV-scFv in terms of colloidal properties.

##### XPS Characterization

XPS analysis is a technique that provides chemical information at a depth of 5 nm approximately. In our study, XPS spectra gave proof that there is a chemical difference between NV-scFv-DTPA_0.6_ and NV-scFv. Regardless of the presence of Na, Cl, P, K, and Si elements (i.e., due to the PBS buffer), component quantification showed a significant increase in the N/Fe ratio on NV-scFv-DTPA_0.6_ compared to NV-scFv (Table 1). As the atomic iron percentage in NV-scFv-DTPA_0.6_ is lower, and a sulfur contribution was revealed on the XPS survey (Appendix A), it is possible to conclude that a nitrogen-rich molecule containing sulfur was conjugated onto NV-scFv, suggesting the presence of DTPA on NV-scFv-DTPA_0.6_.

In addition, thanks to high-resolution XPS spectra, the following statements can be made for both materials. Firstly, the same oxidized iron form was confirmed through a Fe2p_3/2_ component around 710.4 eV (3^+^ environment type) and its corresponding O1s contribution at 529.6 eV. However, the proportion of the O1s component was lower in NV-scFv-DTPA_0.6_, giving proof of another component that was grafted onto NV-scFv. Secondly, the same environment for N since the N1s fitted spectrum leads to one component located at 399.9 eV with the same FWHM (Full Width at Half Maximum) for both NV-scFv and NV-scFv-DTPA_0.6_. This similarity could be explained by the overwhelming quantity of N in scFvs compared to that in DTPA. Lastly, a change in the C1s fitted spectrum with a lower proportion of CH_2_–CH_2_ bonds (284.7 eV) for NV-scFv-DTPA_0.6_ was observed. This could be explained by the masking effect of DTPA-grafting on the C–C bonds of the maleimide groups, and the fact that such grafting did not provide any CH_2_–CH_2_ bonds.

To sum up, there is a distinct possibility that the DTPA-derivatization was successfully performed. Nevertheless, it is difficult to come up with a more precise conclusion on the chemical structures of these NPs using XPS, since the carbon environments are quite identical in NV-scFv and DTPA-SH (i.e., COOH, N–C=O).

##### Physico-Chemical Properties Characterization

Table 2 resumed the colloidal properties including the D_H_ and ζ of all the synthesized NV-scFv with or without DTPA. Regardless of the molar DTPA/Fe ratio used in the synthesis, all evaluated NV-scFv-DTPA showed a size around 80 nm with a highly monodispersed population (PDi < 0.2), and slightly negative charges that allowed further complexation with cationic polymers and siRNA for NM-scFv formulation. All these characteristics are suitable for IV administration. Compared to the non-DTPA derivatized NV-scFv, no significant change in size or charge was recorded for all batches of NV-scFv-DTPA using different DTPA/Fe ratios. These results indicated that the DTPA introduction onto the NV’s surface did not influence the physicochemical properties of our nanocarrier.

Besides, the number of conjugated targeting ligands for each batch of NV-scFv-DTPA was also determined to make sure that the DTPA introduction did not discard the grafted scFv moieties on the NV-scFv surface. The results (Table 2) confirmed our hypothesis that by grafting DTPA at the last step of the synthesis, this surface modification did not influence the number of targeting ligands, which was around 26 scFv ligands/NV for all batches of NV-scFv and NV-scFv-DTPA.

### 3.2. Selection of Appropriate DTPA/Fe Ratio for Efficient Radiolabeling of NM-scFv-DTPA

#### 3.2.1. Formulation of NM-scFv-DTPA

As DTPA was grafted onto the NVs’ surface, it might have had a significant influence on the complexation between NVs, siRNA and cationic polymers. This influence may be due to the steric hindrance of DTPA on the NVs’ surface to other components and could be reflected by a change in size or zeta potential of the corresponding NM-scFv-DTPA. Table 3 presents the size and charge of different NM-scFv-DTPA corresponding to each batch of NV-scFv-DTPA. Indeed, the D_H_ was similar (around 100 nm) for all batches of NM-scFv-DTPA except that of the highest DTPA/Fe ratio (NM-scFv-DTPA_0.6_). For NM-scFv-DTPA_0.6_, a significant increase in size was observed. The size of NM-scFv-DTPA_0.6_ increased from 100 nm to 146 nm (around 50 nm), which might be due to the over-presence of DTPA molecules on the NMs surface. To be more specific, we hypothesize that during the NM-scFv-DTPA formulation, siRNA and cationic polymers interpose into the gap between polymer molecules. It explains why siRNA can be well protected in NMs and the size of such NMs is normally decided by the size of NV-scFv [12,14]. Therefore, if DTPA is over-present on the NV-scFv surface, it might hinder this process and a part of siRNA and cationic polymers have to re-organize themselves out of the polymer layer regions, which may lead to an increase in D_H_.

All batches were positively charged but the increasing presence of DTPA on NV-scFv-DTPA was deemed to decrease the zeta potential of the corresponding NM-scFv-DTPA. As the measurement was carried out in NaNO_3_ 0.01 M (pH = 7.5), the presence of anionic carboxylic groups (pKa = 4) from DTPA on the surface can decrease the ζ of the finalized NM-scFv-DTPA. This difference was more noticeable between non-DTPA NM-scFv and NM-scFv-DTPA_0.6_ with the highest DTPA/Fe ratio, in which the zeta potential decreased dramatically from +21.9 ± 4.4 mV to +6.79 ± 0.93 mV. This finding supports the hypothesis that carboxylic functions are present at the surface of NM-scFv-DTPA, which is needed for further radiolabeling.

#### 3.2.2. Complexation of NM-scFv-DTPA with Non-Radioactive ^115^Indium

Currently, the mainly used diagnostic radiometal nuclides for SPECT are gamma-emitters such as ^99m^Tc (t_1/2_ = 6 h) and ^111^In (t_1/2_ = 2.83 d). Preliminary tests of compatibility between our NM-scFv and these radionuclides showed a good stability level between our NM-scFv-DTPA and ^111^In but a problem with stability was observed for ^99m^Tc. As a result, ^111^In was chosen as a radionuclide for our NM-scFv with SPECT imaging. Recently, most radiolabeling strategies for biomedical applications are based on the complexation between radiometal nuclides and hard donor atoms such as oxygen and nitrogen atoms in a complex ligand. Several chelators such as DOTA and DTPA have been developed to provide hard base donor atoms for strong interactions of the central metal and the coordination sphere, enabling the formation of stable complexes with radiometal nuclides. In the case of Indium, DTPA is one of the best candidates due to its high stability, mild conditions (at room temperature) of reaction and short time required for complexation (around 1 h) [28,30]. As the radiolabeling is based on highly stable In^3+^-DTPA chelate, there is no difference in terms of complexation with DTPA between the radioactive isotope ^111^In and the stable isotope ^115^In [31]. Consequently, due to the regulation of the radioactivity in conventional laboratories, the non-radioactive isotope ^115^InCl_3_ was used to establish the labeling protocol and evaluate the physicochemical properties of NM-scFv-DTPA after the complexation with Indium (NM-scFv-DTPA-In).


*(a) Physico-chemical properties of NM-scFv-DTPA-^115^In*


The complexation between NM-scFv-DTPA and ^115^In was carried out according to the protocol described in Section 2.5.1 and identical to what would be used for further in vivo imaging. As the complexation was performed at room temperature at pH = 5.7 and our NM-scFv was stable in these conditions, the complexation is not likely to have any impact on the stability of our NMs.

The physicochemical properties of different batches of NM-scFv-DTPA-^115^In were reported in Table 4. While the size remained acceptable (around 100 nm) for NM-scFv-DTPA_0.03_-^115^In and NM-scFv-DTPA_0.3_-^115^In, a remarkable increase in D_H_ was recorded for NM-scFv-DTPA_0.6_-^115^In from 145.9 ± 3.9 nm to 206.1 ± 10.4 nm, which is not suitable for IV administration (>200 nm). Therefore, NM-scFv-DTPA_0.03_ and NM-scFv-DTPA_0.3_ were selected for the next text radiolabeling with radioactive ^111^InCl_3_. All NM-scFv-DTPA-^115^In were slightly positively charged and suitable for IV administration.

After the complexation, each batch of NM-scFv-DTPA-In was purified by size exclusion chromatography (SEC) with a PD-10 column. In general, there are two purification methods for radiolabeled NPs including size exclusion chromatography (SEC) with a PD-10 column or a G-25 column, or filtration with an appropriate cutoff combined with centrifugation [32]. According to preliminary tests, SEC with a PD-10 column was chosen due to its higher capacity of separation than the G-25 column and its easier application compared to filtration combined with centrifugation. However, one of the most significant challenges in the SEC method with NPs is a change in the hydrodynamic diameters of obtained samples because of the shift in retention volumes caused by the interaction between NPs and the column stationary phase [33]. Table 5 presents the size of NM-scFv-DTPA-^115^In after the purification with the PD-10 column. After purification with the PD-10 column, two populations of NM-scFv-DTPA-^115^In were obtained including a principal population (70%) with appropriate size for IV administration (65.9 nm and 82.4 nm) and another less important population (30%) with non-appropriate size for further in vivo test (720.4 nm and 1851 nm). To exclude the population of NM-scFv-DTPA-^115^In with big sizes, the membrane filtration was performed on a membrane of 0.22 µm. As shown in Table 5, membrane filtration was useful to eliminate the big-size population and with a good filtration yield (around 80%). The finalized NM-scFv-DTPA-^115^In had a size of around 80 nm with a high polydispersity index that was suitable for IV administration.

#### 3.2.3. Radiolabeling Yield of NM-scFv-DTPA by ^111^In^3+^

NM-scFv-DTPA_0.03_ and NM-scFv-DTPA_0.3_ were radiolabeled with ^111^In^3+^ and the radiolabeling efficiency of each NM-scFv-DTPA-^111^In was determined using thin-layer chromatography. For SPECT imaging, it is required that the yield of radiolabeling be superior to 96% for relevant imaging [34,35]. Between these two candidates, only NM-scFv-DTPA_0.3_ satisfied this criterion (98%), whereas NM-scFv-DTPA_0.03_ did not as its yield was negligible (<10%). These results demonstrated that the number of grafted DTPA on the NM-scFv-DTPA_0.03_ was not up to par, but that of NM-scFv-DTPA_0.3_ was sufficient. Moreover, such a high yield of radiolabeling with NM-scFv-DTPA_0.3_-^111^In was similar to that reported in the study of Helbok et al. In this study on the radiolabeling of lipid-based NPs using DTPA-^111^In, the authors obtained a yield of 98% and good imaging potential with SPECT imaging [34]. Henceforth, the NM-scFv-DTPA_0.3_ was chosen as our optimized NM-scFv-DTPA for further tests on small animals. For all following sections, NM-scFv-DTPA is used to indicate NM-scFv-DTPA_0.3_.

### 3.3. In Vitro Potency of Optimized NM-scFv-DTPA for siRNA Active Delivery into EGFR-Positive Cancer Cells

Although the DTPA introduction onto NV-scFv was shown to not have a significant impact on the number of grafted active targeting ligands, there remains a risk that the active targeting properties of conjugated moieties can be negatively influenced due to the steric hindrance of DTPA in the interaction between these ligands and its targets (EGFRs) on cancer cells. As a result, it is necessary to evaluate in vitro the active targeting functionality of these moieties before and after the DTPA-derivatization.

#### 3.3.1. In Vitro Functionality towards EGFR with the ELISA Experiment

The functionality of our targeted NMs-scFv with/without DTPA towards their target EGFR proteins was performed in vitro with the ELISA experiment. As shown in Figure 3, while the non-targeted NM presented no binding affinity towards EGFR (negligible absorbance for the whole range of concentrations), both NM-scFv with or without DTPA exhibited a binding affinity for this protein with increasing absorbance correlated to the NM’s concentration.

Besides, for the whole studied range of concentrations, there was no difference in the absorbance between NM-scFv and optimized NM-scFv-DTPA, indicating the non-significant impact of DTPA introduction on the functionality of the final NMs.

#### 3.3.2. siRNA Protection Capacity of Optimized NM-scFv-DTPA-^115^In

It is required that the optimized NM-scFv-DTPA be always able to protect siRNA and that the stability of the NMs formulation is not affected both during the complexation with indium and the purification process with a PD-10 column. To this end, the optimized NM-scFv-DTPA-^115^In was controlled in terms of its siRNA protection capacity on the agarose gel by the electrophoresis method (Figure 4). Without the added heparin, as the formulation is stable, no fluorescent band of free siRNA was observed. On the contrary, if heparin was added, the NM-scFv-DTPA-^115^In was destabilized due to heparin’s high negative charges and free siRNA was liberated. The fluorescent band corresponding to free siRNA was detected at the same location and intensity as that of the naked siRNA. All these results revealed that after the complexation with indium and the purification with SEC, our optimized NM-scFv-DPTA-^115^In was still able to perfectly protect the encapsulated siRNA.

#### 3.3.3. In Vitro Active siRNA Delivery into EGFR-Positive Non-Small Lung Cancer Cells

Following the binding affinity towards EGFR, the active siRNA delivery of NM-scFv and that of the optimized NM-scFv-DTPA was evaluated and compared in vitro. To this end, the non-small cell lung cancer (NSCLC) cells overexpressing EGFR and luciferase (H460-Luc) were chosen as our cell models. NSCLC is a subtype that comprises the majority (about 85%) of lung cancers and is the leading cause of cancer-related mortality among men and women. Many studies have revealed the close association between EGFR-overexpression (from 40 to 80% cases) and a reduced survival rate, a higher risk of metastasis and poorer chemosensitivity in NSCLC [36,37]. Therefore, the application of EGFR-active targeting with NMs in NSCLC treatment is drawing increasing interest. Among different EGFR-overexpressed NSCLC cell lines, the H460 cell line model is one of the most widely used for in vitro experiments with EGFR-targeting and was proven a good in vitro model for EGFR-targeting in NSCLC [9,38]. As these cancer cells stably overexpressed the luciferase gene, anti-luciferase siRNA was complexed in our nanomedicines. The successful anti-luciferase siRNA delivery with NMs into H460-Luc cells leads to a downregulation in luciferin protein by inhibiting the translation of luciferase mRNA into luciferin protein. This downregulation can be detected with the luminescence assay. As shown in Figure 5, the targeting properties of NM-scFv were confirmed by an enhanced siRNA transfection efficiency level by a factor of 1.7 compared to non-functionalized NM (around 67% vs. 39%, respectively). Furthermore, with the optimized DTPA/Fe ratio = 0.3, the DTPA-derivatization did not show any significant impact on siRNA delivery (*p* < 0.01) and the transfection efficiency remained around 67%. As expected, this efficiency is lower than that of commercialized transfection agent Lipofectamine (67% vs. 81%). However, this transfection agent is not compatible with IV injection and is, therefore, used only as our positive control. Our transfection efficiency is relatively good compared to that reported in other studies in the field of siRNA delivery with non-viral delivery systems. To the best of our knowledge, the transfection efficiency of current transfection agents varied from 32% to 85% [14]. All previous results demonstrated that at the optimized DPTA/Fe ratio equivalent to 0.3, there is no impact of the DTPA-derivatization on the active delivery of siRNA between NM-scFv and the optimized NM-scFv-DTPA.

### 3.4. Stability Control of the Optimized Nanomedicine in In Vivo Conditions

#### 3.4.1. Radiolabeling of the Optimized NM-scFv-DTPA with ^111^In

The selected NM-scFv-DTPA was radiolabeled with ^111^In and purified with a PD-10 column. During the purification, the elution profile of the purified samples was established by measuring both the NIRF intensity and radioactivity of each fraction (180 µL/fraction). The six most concentrated fractions (the fourth to ninth fractions) were mixed together. At these fractions, elution profiles established with both NIRF intensity and radioactivity were shown to be overlaid (Appendix A). This superposition demonstrated that most radionuclides were complexed with our optimized NM-scFv-DTPA. Afterward, the pooled radio-labeled NM-scFv-DTPA-^111^In was filtered through a membrane of 0.22 µm as described in Section 3.2.2. The finalized NM-scFv-DTPA-^111^In had a pH = 7.4 ready for IV injection.

To determine the concentration of the injected NM-scFv-DTPA-^111^In, its NIRF was measured and the concentration of the injected samples was determined to be around 782 nM in siRNA (10.4 µg/mL or 104 mg of iron/L) using the calibration range previously established (Appendix A). The dose of NMs used for both NIRF and SPECT imaging was fixed at 152 µg of siRNA/kg of mice per injection. In gene therapy with siRNA, different doses of siRNA can be used for in vivo experiments and may vary from hundreds of µg to hundreds of mg/kg of mice depending on the target protein, the delivery system, the administration way, and whether chemotherapy is involved or not [39]. In general, the delivery of siRNA with an appropriate delivery system may help not only to decrease the required dose significantly but also to minimize the risk of side effects. Especially, in the delivery with active targeting systems and in combination with chemotherapy, this dose can be reduced to 150 µg of siRNA/kg of mice as shown in the study of Lander et al., or 200 µg of siRNA/kg in the study of Sun et al., [11,40].

#### 3.4.2. Stability Control of Radiolabeled NM-scFv in In Vivo Conditions Using NIFR Imaging

^111^In-radiolabeled and DTPA-derivatized NM-scFv made of control siRNA fluorescent-labeled with Atto^TM^740 were IV injected into healthy mice, at a dose of 152.5 µg of siRNA/kg of mice, for the stability control of our optimized nanosystem in in vivo conditions. Ex vivo images of two fluorescence signals from Dylight^TM^680 (on NV-scFv-DTPA) and Atto^TM^740 (on siRNA) were used in this study (Figure 6).

Firstly, the signal of Dylight^TM^680 showed a strong accumulation of our NV-scFv-DTPA in the liver without important fixation in the spleen, 40 min after injection.

The signal of Atto^TM^740 on siRNA was, as expected, weaker than that of Dylight^TM^680 due to its lower quantum efficiency (approximately by a factor of 40). At 40 min after injection, a similar biodistribution level of siRNA to that of NV-scFv was revealed with the main liver accumulation and marked elimination at the digestive level.

These results showed that at the same time of experiments, the biodistribution of siRNA is strictly identical to that of NV-scFv, which reflects good stability in in vivo conditions of the complexation between NV-scFv and siRNAs despite their relatively weak electrostatic forces.

### 3.5. 3D In Vivo Tracking with SPECT-CT Imaging

The current study was carried out to develop and validate a radiolabeling method for our EGFR-targeting nanomedicine for its tracking in in vivo conditions. Therefore, one of the main objectives is to successfully track our finalized NM-scFv-DTPA-^111^In with SPECT-CT imaging. To validate our proof of concept with such NMs, in vivo studies were first performed with healthy mice. To this end, our optimized ^111^In-radiolabeled NM-scFv-DTPA (NM-scFv-DTPA-^111^In) at 3.7 ± 0.74 MBq was IV injected into healthy mice at a dose of 152 µg of siRNA/kg of mice and SPECT-CT images were acquired at 40 min after injection (Figure 7A). All animals were sacrificed at the time of 40 min post-injection in accordance with preliminary results. Indeed, our preliminary results showed that from the time of 40 min post-injection, the biodistribution of our nanosystem does not change anymore due to a lack of stealthiness of our nanosystem that is subjected to our further studies. Finally, the radioactivity accumulated in organs of interest including the liver, heart, and lung was quantified and compared to the total radioactivity injected (Figure 7B).

In vivo tracking images of our NM-scFv-DTPA-^111^In at 40 min after injection showed a weak residual activity at the heart level and an intensive hepatic fixation with moderate intestinal elimination. Besides, the negligible renal radioactivity, the absence of urinary elimination, and the absence of radioactivity in bone marrow demonstrated the excellent in vivo stability of radiolabeling with ^111^In [41]. The radioactivity distribution of the labeled NMs within the liver, heart, and lung was determined to be 32.0 ± 2.5%, 0.7 ± 0.3%, and 2.8 ± 0.6%, respectively. These in vivo distribution values were calculated based on appropriate volumes of interest (VOIs) and included the radioactivity of the tissue and the blood pool of the organ, providing an estimate of organ accumulation [41]. This predominant accumulation of NMs in the liver shows a lack of stealthiness in our nanomedicines. Especially, the impact of the formation of a protein corona around our NM scFv in in vivo conditions that may enhance the NM-scFv clearance via the mononuclear phagocytic system will be further clarified in our upcoming study [42]. In addition, our upcoming study will be performed in tumorized mice where the overexpression of EGFR in tumors could improve the accumulation of our targeted nanomedicines. All the previous results clearly validated our concept on the possibility of tracking our NM-scFv-DTPA-^111^In with SPECT-CT imaging, which is interesting for future experiments on tumor-bearing mice. Moreover, in addition to non-invasive properties, SPECT-CT may facilitate long-term studies of cancer progression without sacrificing animals at different tumor stages. This advantage is highly remarkable compared to other molecular imaging techniques [43].

## 4. Conclusions

The development and the quality control of a radiolabeling method for our EGFR-targeted NM-scFv loading siRNA were presented in this paper. Our EGFR-targeted nanomedicines with anti-EGFR scFv ligands (NM-scFv) were able to not only effectively protect and deliver siRNA into EGFR-overexpressing cancer cells but also can be tracked in vivo with single-photon emission computed tomography coupled with computed X-tomography (SPECT-CT) imaging. The in vitro studies showed that our optimized DTPA-derivatized NMs were able to perfectly protect the formulated siRNA and preserve their active targeting properties, as reflected by a 1.7-fold higher siRNA transfection efficiency level in EGFR-positive cancer cells compared to non-targeted NMs. Furthermore, NIRF ex vivo images have demonstrated good stability in in vivo conditions of our formulation between NV-scFv and siRNA despite their weak electrostatic forces. High radiolabeling yield, and excellent in vivo radiolabeling stability were also achieved with our optimized NM-scFv using the DTPA chemistry.

To conclude, a radiolabeling method for our EGFR-targeted nanomedicine was developed and controlled for quality in this study, enabling in vivo tracking with nuclear imaging. Furthermore, this radiolabeling method plays a vital role in the next step of the evaluation of their in vivo biological activities and/or histological study. For further studies, our NMs-scFv-DTPA will be used to deliver siRNA with therapeutic activities such as BcL-xL or survivin siRNA and have their active targeting properties and safety tested in vivo.

## Figures and Tables

**Figure 1 pharmaceutics-14-02679-f001:**
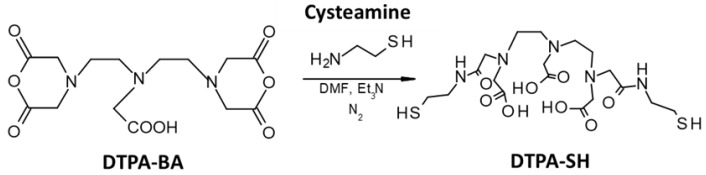
Thiolation reaction of DTPA.

**Figure 2 pharmaceutics-14-02679-f002:**
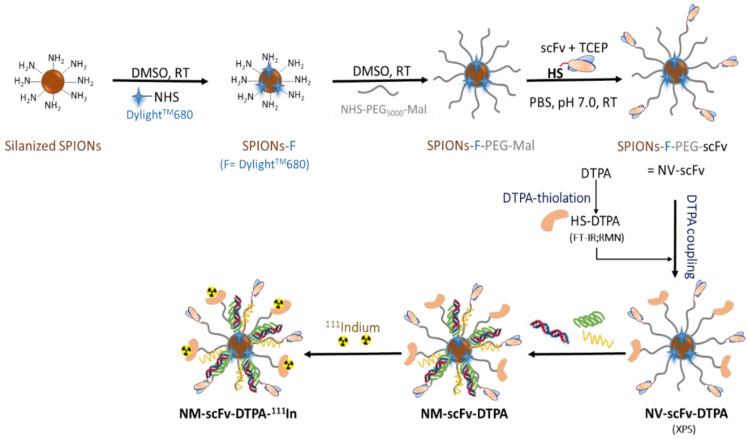
Schematic representation of NM-scFv-DTPA-^111^In synthesis.

**Figure 3 pharmaceutics-14-02679-f003:**
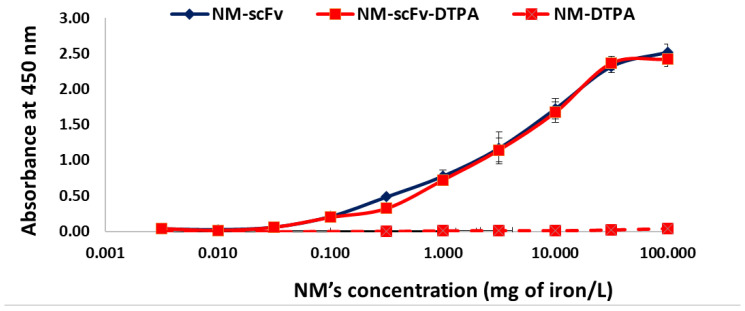
Functionality interpreted by the absorbance at 450 nm of PpL’s substrate obtained from the ELISA experiment of NM-scFv, optimized NM-scFv-DTPA and corresponding NM-DTPA regarding EGFR.

**Figure 4 pharmaceutics-14-02679-f004:**
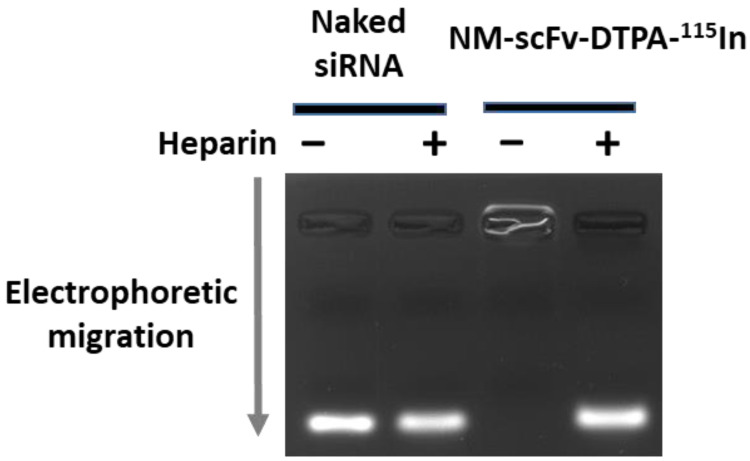
Gel retardation assay to detect free siRNA in the optimized NM-scFv-DTPA-^115^In (DTPA/Fe molar ratio: 0.3) with (+) or without (−) heparin.

**Figure 5 pharmaceutics-14-02679-f005:**
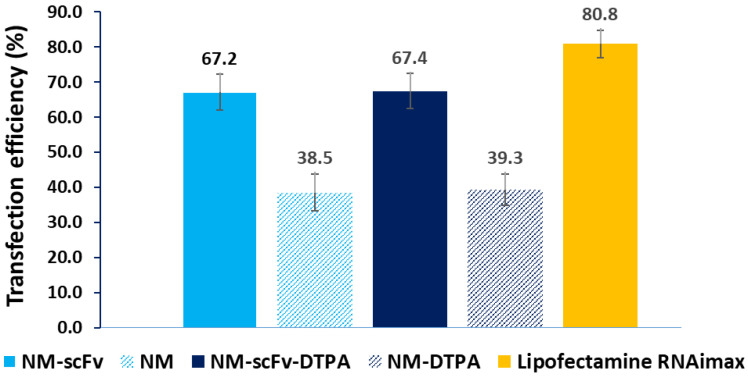
Anti-Luciferase siRNA transfection efficiency of different NM-scFv and the optimized NM-scFv-DTPA into H460-Luc cancer cells.

**Figure 6 pharmaceutics-14-02679-f006:**
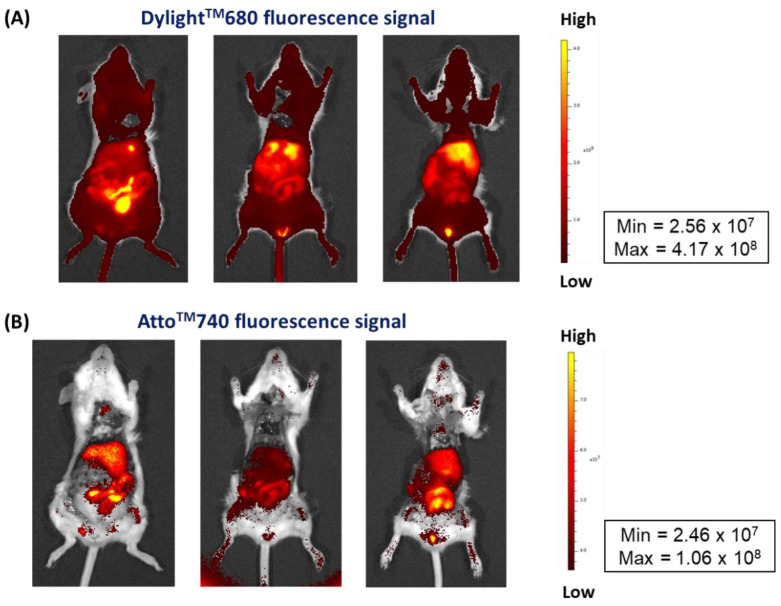
Ex vivo NIRF images of the optimized ^111^In-radiolabeled and DTPA-derivatized NM-scFv made of control siRNA fluorescent-labeled with Atto^TM^740 (DTPA/Fe molar ratio: 0.3) at 40 min after injection. (**A**) NV-scFv-DTPA signal, (**B**) siRNA signal.

**Figure 7 pharmaceutics-14-02679-f007:**
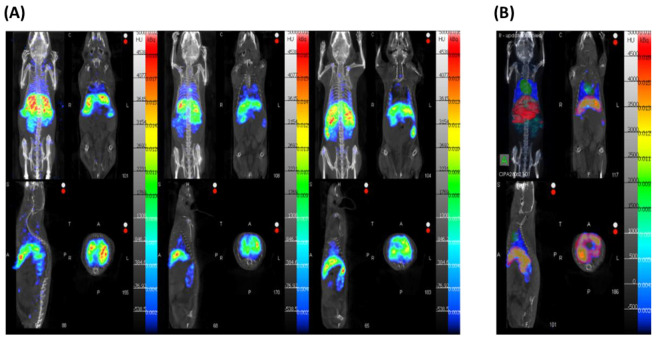
In vivo SPECT/CT images of the optimized NM-scFv-DTPA-^111^In (DTPA/Fe molar ratio: 0.3) (**A**) The 3D reconstructed and co-registered SPECT and CT images are shown together with coronal, sagittal, and axial images (from left to right, up to down). (**B**) Quantification of NMs accumulation by liver, heart, and lungs with VOIs.

**Table 1 pharmaceutics-14-02679-t001:** XPS quantification (at. %) on NV-scFv and NV-scFv-DTPA_0.6_ sample drops.

Batches	C	O	N	Fe	Na	Cl	P	K	Ca	Si	N/Fe
**NV-scFv**	42.7	38.1	4.2	4.1	4.5	1.6	2.7	0.5	-	1.6	1.0
**NV-scFv-DTPA_0.6_**	43.0	38.2	3.4	1.9	5.5	1.5	4.5	0.4	0.7	0.9	1.8

**Table 2 pharmaceutics-14-02679-t002:** Physico-chemical properties and number of anti-EGFR scFv ligands on different batches of NV-scFv-DTPA.

Batch	DTPA/Fe Ratio	D_H_ (nm)	PDi	ζ (mV)	Number of scFv/NV
NV-scFv	0	75.4 ± 2.0	0.17 ± 0.01	−3.5 ± 1.0	26 ± 3
NV		78.8 ± 0.2	0.16 ± 0.01	−4.9 ± 1.1	0
NV-scFv-DTPA_0.03_	0.03	85.0 ± 2.1	0.18 ± 0.01	−3.3 ± 0.3	24 ± 2
NV-scFv-DTPA_0.3_	0.3	75.6 ± 1.3	0.16 ± 0.01	−2.9 ± 0.6	28 ± 2
NV-scFv-DTPA_0.6_	0.6	79.9 ± 1.5	0.16 ± 0.01	−4.4 ± 0.9	24 ± 2

**Table 3 pharmaceutics-14-02679-t003:** Physico-chemical properties of NM-scFv made of different NV-scFv-DTPA.

Batch	DTPA/Fe Ratio	D_H_ (nm)	PDi	ζ (mV)
NM	0	99.0 ± 2.5	0.27 ± 0.01	+15.3 ± 1.3
NM-scFv	0	100.0 ± 3.5	0.24 ± 0.03	+21.9 ± 4.4
NM-scFv-DTPA_0.03_	0.03	92.4 ± 5.0	0.21 ± 0.01	+18.5 ± 2.6
NM-scFv-DTPA_0.3_	0.3	91.0 ± 4.7	0.23 ± 0.01	+12.5 ± 3.4
NM-scFv-DTPA_0.6_	0.6	145.9 ± 3.9	0.27 ± 0.01	+6.79 ± 0.93

**Table 4 pharmaceutics-14-02679-t004:** Physico-chemical properties of NM-scFv after the complexation with InCl_3_.

Batch	DTPA/Fe Molar Ratio	D_H_ (nm)	PDi	ζ (mV)
NM-scFv-DTPA_0.03_-^115^In	0.03	103.0 ± 9.1	0.27 ± 0.01	+0.7 ± 0.3
NM-scFv-DTPA_0.3_-^115^In	0.3	113.2 ± 0.5	0.24 ± 0.01	+2.0 ± 1.8
**NM-scFv-DTPA_0.6_-^115^In**	0.6	**206.1 ± 10.4**	**0.21 ± 0.02**	**+0.3 ± 0.1**

**Table 5 pharmaceutics-14-02679-t005:** Hydrodynamic size of different batches of NM-scFv-DTPA-^115^In before and after membrane filtration.

Batch	Before Filtration		After Membrane Filtration		Filtration Yield
	D_H_ (nm)	PDi	D_H_ (nm)	PDi	
NM-scFv-DTPA_0.03_-^115^In	65.9 ± 3.3 (70%) 720.4 ± 17.5 (30%)	0.42 ± 0.01	77.6 ± 3.5 (100%)	0.24 ± 0.01	81.5%
NM-scFv-DTPA_0.3_-^115^In	82.4 ± 2.6 (70%) 1851 ± 326.1 (30%)	0.53 ± 0.01	85.7 ± 0.6 (100%)	0.23 ± 0.01	77.8%

## Data Availability

The data presented in this study are available in this article.

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
