# Peer review of "Radiolabeling, Quality Control and In Vivo Imaging of Multimodal Targeted Nanomedicines"

_pharmaceutics, 2022, doi:10.3390/pharmaceutics14122679_

Round 1

Reviewer 1 Report

The authors have established the production of EGFR-targeted nanomedicine (NM-scFv-218 DTPA) in order to make it ready for readers authors must adress some points:

Authors must clarify all the abbreviations used in the text for the time its appear, like: NHS-PEG5000-Maleimide... and many others...

Authors must include a paragraph in the introduction about 111In and it applicability

Why authors used PBS as medium for Dh, PDi and Zeta potential? 

In section 2.5.1. Preparation and physico-chemical characterizations of NM-scFv-DTPA-115In authors must describe the most probable mechanism of radiolabeling with 111In.  Please check the reference: https://link.springer.com/article/10.1186/s41181-022-00161-4

In section 2.7. Small-animal in vivo experiments:  how many animals were used? 

In section 2.7.2. Stability control of optimized NM-scFv-DTPA-111In in in vivo conditions: What was de activity used? How  animals were sacrificed? 

In section 2.7.3. Small-animal SPECT-CT imaging: What was the via used for administration? What was the activity of 11In used? How many animals were used in the assay? Why authors sacrificed the animals in the time of  40 min ? Why not wait 2 hours?

In section 3.1.2. DTPA functionalization of NV-scFv author must improve the discussion, see the reference: https://link.springer.com/article/10.1007/s11095-017-2320-2

In section 3.5. 3D in vivo tracking with SPECT-CT imaging: Author must explain the high uptake in liver, heart, and lungs.

Authors must perform the following assays, prior the publication.

Partition coefficient study

Plasma stability 

Binding study

Cell proliferation 

Reviewer 2 Report

This article puts forward a radiolabeling method for EGFR-targeted nanomedicine, which is developed and controlled for their quality enabling tracking with nuclear imaging in vivo. It is a topic of interest to the researchers in the related areas but the paper needs improvement before acceptance for publication. My detailed comments are as follows:

1) NM-scFv is based on the electrostatic complexation of targeted nanovector (NV-scFv), siRNA and two cationic polymers. It contains a variety of non-inorganic materials and has the risk of bacterial infection. So, the authors should describe the colony control and long-term storage stability of nanomedicine.

2) 2.3.1, the yield of Thiolation of DTPA-BA should be given.

3) Table 3 and 4, why did the zeta potential of NM-scFv-DTPA0.03 and NM-scFv-DTPA0.3 decrease significantly after complexation with In3+, a cation?

4) Figure 3, error bars are required for each data point.

5) What about the in vivo toxicity and pharmacokinetic studies of NM-scFv?

Round 2

Reviewer 1 Report

Authors have completed most of the comments. However is necessary to add the following information in the text:

2.6. Radiolabeling efficiency for determination of the suitable DTPA/Fe for in vivo SPECT- 312 CT imaging

Must include the activity of radionuclide used in the radiolabeling process.  

2.7. Small-animal in vivo experiments

Must include the number of animals per group.

Must include the activity of radionuclide injected per/animal 

2.7.1. Radiolabeling of NM-scFv-DTPA with 111InCl3

Must include the activity of radionuclide used

2.7.3. Small-animal SPECT-CT imaging 

Must include the number of animals used per group

Must include the activity of radionuclide injected per animal

Must include the explanation given in the reviewer response regarding the time used.

Author Response

Thank you for your useful comments that will absolutely improve the quality of our paper. After careful consideration, we would like to answer your comments, as follow:

2.6. Radiolabeling efficiency for determination of the suitable DTPA/Fe for in vivo SPECT-CT imaging

Must include the activity of radionuclide used in the radiolabeling process. 

Information added at line 317: “Two batches of NM-scFv-DTPA at DTPA/Fe= 0.03 and 0.3 were complexed with 111InCl3 at a radioactivity of 138.26 ± 9.51 MBq and evaluated in terms of radiolabeling efficiency”.

2.7. Small-animal in vivo experiments

Must include the number of animals per group.

Information added at line 329: “Female Balb/C nude mice (three animals per group for two groups) were purchased from Janvier Laboratories with a weight around 20 g/individual (7 weeks)”.

Must include the activity of radionuclide injected per/animal.

This information was added at line 364-365: “The 111In-labeled NM-scFv was IV injected into three mice at a dose of 152.5 µg of siRNA per kg of mouse at 3.7 ± 0.74 MBq in radioactivity.

2.7.1. Radiolabeling of NM-scFv-DTPA with 111InCl3

Must include the activity of radionuclide used.

Information added at line 339: “The protocol of complexation between NM-scFv-DTPA and 111InCl3 was identical to that described in 2.5.1 at an initial radioactivity of 138.26 ± 9.51 MBq.”

2.7.3. Small-animal SPECT-CT imaging 

Must include the number of animals used per group.

Must include the activity of radionuclide injected per animal.

Information added at lines 364-365: “The 111In-labeled NM-scFv was IV injected into three mice at a dose of 152.5 µg of siRNA per kg of mouse at 3.7 ± 0.74 MBq in radioactivity.”

Must include the explanation given in the reviewer response regarding the time used.

Information added at lines 722-726 in 3.5. 3D in vivo tracking with SPECT-CT imaging: “All animals were sacrificed at the time of 40 minutes post-injection in accordance with preliminary results. Indeed, our preliminary results showed that from the time of 40 minutes post-injection, the biodistribution of our nanosystem does not change any-more due to a lack of stealthiness of our nanosystem that is subjected to our further studies.

Reviewer 2 Report

The authors have addressed my issues. I've no further comments now.

Author Response

Thank you very much for your comments.